# Radiochemical Feasibility of Mixing of ^99m^Tc-MAA and ^90^Y-Microspheres with Omnipaque Contrast

**DOI:** 10.3390/molecules27217646

**Published:** 2022-11-07

**Authors:** Chang-Tong Yang, Pei Ing Ngam, Vanessa Jing Xin Phua, Sidney Wing Kwong Yu, Gogna Apoorva, David Chee Eng Ng, Hian Liang Huang

**Affiliations:** 1Department of Nuclear Medicine and Molecular Imaging, Radiological Sciences Division, Singapore General Hospital, Outram Road, Singapore 169608, Singapore; 2Duke-NUS Medical School, 8 College Road, Singapore 169857, Singapore; 3Department of Diagnostic Imaging, National University Hospital Singapore, 5 Lower Kent Ridge Road, Singapore 119074, Singapore; 4Department of Vascular and Interventional Radiology, Radiological Sciences Division, Singapore General Hospital, Outram Road, Singapore 169608, Singapore

**Keywords:** radiochemical purity, ^90^Y microspheres, Technetium-99m macroaggregated albumin(^99m^Tc-MAA), ^90^Y-SIRT radioembolization, omnipaque contrast

## Abstract

Yttrium-90 (^90^Y) microspheres are widely used for the treatment of liver-dominant malignant tumors. They are infused via catheter into the hepatic artery branches supplying the tumor under fluoroscopic guidance based on pre-therapy angiography and Technetium-99m macroaggregated albumin (^99m^Tc-MAA) planning. However, at present, these microspheres are suspended in radiolucent media such as dextrose 5% (D5) solution. In order to monitor the real-time implantation of the microspheres into the tumor, the ^90^Y microspheres could be suspended in omnipaque contrast for allowing visualization of the correct distribution of the microspheres into the tumor. The radiochemical purity of mixing ^90^Y-microspheres in various concentrations of omnipaque was investigated. The radiochemical purity and feasibility of mixing ^99m^Tc-MAA with various concentrations of a standard contrast agent were also investigated. Results showed the radiochemical feasibility of mixing ^90^Y-microspheres with omnipaque is radiochemically acceptable for allowing real-time visualization of radioembolization under fluoroscopy.

## 1. Introduction

Yttrium-90 selective internal radiation therapy (^90^Y-SIRT) is a form of liver-directed internal radiotherapy used in the treatment of unresectable hepatocellular carcinoma (HCC) and colorectal liver metastasis [1,2,3,4,5,6]. ^90^Y-SIRT uses high-energy beta particle-emitting radioisotope microspheres to embolize the tumor that is predominantly supplied by the hepatic artery with relative sparing of the normal liver parenchyma, mainly supplied by the portal veins [7,8,9]. Comprehensive multidisciplinary collaboration of nuclear medicine, interventional radiology, and medical oncology is, therefore, crucial in maximizing the efficacy and minimizing the adverse effects of SIRT [10,11,12]. ^90^Y-SIRT is a two-stage procedure: (1) pre-SIRT mapping, and (2) SIRT administration [13]. At the pre-SIRT stage, the selected patient undergoes a catheter angiogram to scrutinize the arterial supply of the targeted tumor(s), followed by radionuclide mapping using Technetium-99m macroaggregated albumin (^99m^Tc-MAA) [14]. Injection of iodinated contrast into a microcatheter that is placed in the selected hepatic artery allows delineation of the arterial anatomy and perfused hepatic territory, as well as detection of any reflux of contrast prior to administration of ^99m^Tc-MAA. At the SIRT treatment stage, ^90^Y-SIRT is administered according to the microcatheter positions at the pre-SIRT stage. As recommended by the manufacturer, the “sandwich” technique that precludes direct contact of iodinated contrast with the microspheres can be used to confirm the flow of the particles during the administration. ^99m^Tc-MAA that is of comparable size to the ^90^Y-microspheres is used to simulate the distribution of microspheres and provides details on the degrees of radiation to the tumor, healthy liver, and extrahepatic organs [15,16,17,18].

Despite the ideal radiochemical properties of ^99m^Tc-MAA, there are a few technical factors that may potentially results in unexpected implantation of ^99m^Tc-MAA and ^90^Y-SIRT. During most of the embolization procedures, the radio-opaque contrast is used to visualize the flow of the embolic particles during administration. Unfortunately, both ^99m^Tc-MAA and ^90^Y-SIRT are radiolucent; therefore, their real-time administration is purely guided by visual estimation of the position of the microcatheter. For ^99m^Tc-MAA administration, there can be reflux of the particles if the hand injection is performed too rapidly, or streaming of particles with uneven distribution in some of the branches of hepatic artery if the hand injection is performed too softly. On the other hand, there can be angiographic stasis during ^90^Y-SIRT administration, especially when large numbers of particles are used [19,20]. In addition, both ^99m^Tc-MAA and ^90^Y-SIRT are administrated using different delivery systems. ^99m^Tc-MAA is delivered using a 3 mL syringe connected directly to the microcatheter, while ^90^Y-SIRT is delivered via a long tubing and a 20 mL syringe. This results in a difference in pressure generated at the tip of the microcatheter and, hence, a difference in pressure and the number of particles injected along the microcatheter, as well as potential subtle shift in the position of the microcatheter during administration. This may lead to discordant radiotracer distribution on both ^99m^Tc-MAA mapping and ^90^Y-SIRT scan, which has implications for pre-SIRT dosimetry planning and post-SIRT adverse effects. A proposal to infuse ^99m^Tc-MAA or ^90^Y microspheres within standard contrast agents is one possible solution. However, it is unknown whether ^99m^Tc-MAA or ^90^Y microspheres will remain stable and effective in such contrast agents. Therefore, we investigate the ex vivo the compatibility and stability characteristics of ^99m^Tc-MAA and ^90^Y microspheres suspended in a contrast agent (omnipaque), in order to allow real-time visualization and monitoring of the microparticle distribution during the implantation procedure.

## 2. Results and Discussion

^99m^Tc-MAA has been used for radionuclide mapping in the pre-SIRT stage. ^99m^Tc-MAA comparable to the size of the ^90^Y-microspheres was used to simulate the distribution of ^90^Y-microspheres to provide details on the degrees of radiation to the tumor, healthy liver, and extrahepatic organs. The standard procedure involves a fine catheter being placed at the root of the artery supplying the region meant to be treated. A small bolus of contrast such as omnipaque is injected to allow delineation of the arterial anatomy and, in cases where intra-arterial CT is available, to allow a CT scan of the area of liver perfused by the artery to be performed. During contrast bolus injection, feedback of the flow of the contrast can be obtained by intermittent fluoroscopy [8]. ^99m^Tc-MAA is then injected into the catheter when the catheter position is deemed satisfactory; however, no feedback on the flow of contrast can be obtained as ^99m^Tc-MAA is suspended in a radiolucent liquid. In order for real-time monitor implantation of microspheres into the tumor, ^99m^Tc-MAA particles are mixed in omnipaque solution.

The radiochemical purity and feasibility of mixing ^99m^Tc-MAA particles with various omnipaque solutions were investigated. As shown in Table 1, the average radiochemical purity of ^99m^Tc-MAA before the addition of omnipaque was 99.83% ± 0.068%. The radiochemical purity of ^99m^Tc-MAA at 4 h after the addition of omnipaque and at room temperature was 98.73 ± 0.578%. There was a slight decrease of 1.1% in the radiochemical purity of ^99m^Tc-MAA. However, average ^99m^Tc-MAA radiochemical purity of 98.73% is well above the passing limit of 90%. The results showed that ^99m^Tc-MAA could be safely suspended in a solution of omnipaque, which would improve visualization of the flow of ^99m^Tc-MAA particles in real time. The implication would be a safe injection of ^99m^Tc-MAA particles with better pre-therapy planning of ^90^Y radioembolization.

The radiochemical purities of mixing ^90^Y-microspheres in different concentrations of omnipaque diluted with D5 or normal saline were then investigated to see if they were acceptable in terms of stability of the ^90^Y-microspheres, leaching of free ^90^Y from the microspheres, or creating potential byproducts. The radiochemical purities of the ^90^Y-microsphere samples were measured using iTLC with iTLC silica gel (1 × 5 cm) as the stationary phase and normal saline as the mobile phase. The radiochemical purities of the ^90^Y-microsphere samples were calculated on the basis of the integral areas of two peaks of ^90^Y-microsphere and free ^90^Y leaching from microspheres in the iTLC chromatogram (Figure 1). The experiments were repeated three times, and the average results are reported in Table 2. The proportional integration of ^90^Y-microsphere activity was 98.08%, 95.82%, 97.65%, 95.96%, and 95.67%, in samples **1**–**5**, respectively. Thus, leaching of ^90^Y from the microspheres was less than 5% for most samples. Radiochemical purities on iTLC ranged from 94.96%, when omnipaque was diluted to 75% with normal saline, to 97.65%, when omnipaque was diluted to 50% with D5.

It should be noted that different methods ertr used for assessing the radiochemical purity of ^99m^Tc-MAA and ^90^Y-microsphered in this study. For ^99m^Tc-MAA, the iTLC strips were cut into yeo halves, with the radioactivity in each half representing ^99m^Tc-MAA and free ^99m^Tc determined using a multichannel analyzer. For ^90^Y-microspheres, the multichannel analyzer could not be used to measure the radioactivity of ^90^Y-microspheres and free ^90^Y leaching from microspheres due to its pure beta-emitter of ^90^Y isotope with a decay energy of 0.94 MeV and no gamma emission [21,22].

All five samples of ^90^Y-microspheres **1**–**5** further underwent microscopic imaging (Figure 2) after suspension of ^90^Y-microspheres in omnipaque contrast. The microscopic images showed favorable mono-dispersion of ^90^Y-microspheres. The suspension of ^90^Y-microspheres in omnipaque did not cause aggregation of microspheres, which may have caused aberrant deposition of microspheres. The current standard of care “sandwich” technique involves omnipaque injection just before and just after ^99m^Tc-MAA or ^90^Y-sphere infusion, which is not noted to have an increased risk of complications such as blood clots. Thus, it would be unlikely that the combination of ^99m^Tc-MAA or ^90^Y-microspheres with omnipaque would result in such complications and, even if so, these are unlikely to be more than the current standard of care. However, this may need further evaluation with relevant animal models for in vivo studies if necessary. The combination of the ^99m^Tc-MAA or ^90^Y-microspheres with omnipaque did not increase the radiolysis of ^99m^Tc-MAA or ^90^Y-microspheres, as radiolysis depends on the linear energy transfer (LET) of the radioisotopes [23]. After adding contrast omnipaque, the radiolysis should not increase. Moreover, ^99m^Tc is low-energy gamma particle; hence, the radiolysis of ^99m^Tc-MAA should be much lower than that of ^90^Y-microspheres. We tested the stability of ^99m^Tc-MAA by adding contrast omnipaque at the same volume of serum incubated at 37 °C. TLC chromatograms at 30 min and 3 h showed that ^99m^Tc-MAA is stable in contrast to serum (Appendix A). Moreover, if radiolysis or in vivo breakdown occurs, the free ^99m^Tc would localize in the thyroid and stomach. This did not happen in our study.

Due to its comparable size, ^99m^Tc-MAA has been used to simulate the distribution of ^90^Y-microspheres and pre-SIRT mapping for ^90^Y-microspheres to provides details on the degrees of radiation to the tumor, healthy liver, and extrahepatic organs [24]. During most embolization procedures, the interventional radiology is able to see exactly what is going on during the delivery of the embolic because it is radio-opaque. The current ^99m^Tc MAA and ^90^Y-microsphere administrations are performed “blind” because both are radiolucent. If the suspension of ^99m^Tc MAA and ^90^Y-microspheres in contrast is possible, this would allow for interventional radiologists and injection nuclear medicine physicians to monitor and adjust the injection in terms of catheter position or pressure in order to achieve the intended distribution of the ^90^Y-microspheres.

To the best of our knowledge, there has been no published data on the radiochemical feasibility of mixing ^99m^Tc-MAA and ^90^Y-microspheres with omnipaque contrast.

## 3. Materials and Methods

The radioactivity measurement of ^90^Y microspheres was performed using an AtomLab^TM^ 500 dose calibrator from Biodex Medical System, Inc., New York, USA. ^90^Y microspheres were purchased from SIRTeX Medical Europe GmbH, Bonn, Germany. Omnipaque contrast medium (350 mgI/mL iohexol) was purchased from GE healthcare. G5-5% Glucose Intravenous Infusion B.P (D5) and 0.9% sodium chloride injection B.P (saline solution) were purchased from B. Braun Medical Industries S/B. Thin-layer chromatography (TLC) silica gel 60 F_254_ aluminum sheets (20 × 20 cm) were purchased from Merck KGaA, and cut into 2 × 10 cm strips. The radioactivity measurement of ^99m^Tc-MAA on the TLC strips was determined using an MCA-3 Series/P7882 multichannel analyzer from FAST ComTec GmbH. Instant thin-layer chromatography (iTLC) was performed using a Bioscan AR-2000, Wilmington, MA, USA with a P10 cylinder containing methane and argon gas, A Millex^®^ GS filter unit 0.22 μm MF-Millipore^TM^ MCE membrane was purchased from Merck Millipore Ltd. Ground-edge microscopic slides (1″ × 3″, 1–1.2 mm thickness) from Sail Brand and prewashed borosilicate D 263M glass (22 × 22 mm) of hydrolytic class I Deckgläser from Menzel-Gläser as the cover glass were used to place a drop of ^90^Y-microsphere sample for imaging under a microscope. All microscopic imaging was performed using an Olympus BX40 Manual Clinical Microscope, Shinjuku-ku, Tokyo, Japan with an objective of 10×/0.25 and a 12 V white LED light source. Commercially available kits of TechneScan LyoMAA from Mallinckrodt Medical B.V., Dublin, Ireland were used for labeling ^99m^Tc. Each kit contains 2 mg of macro aggregated human serum albumin particles; 95% of the particles in each vial ranged between 10 and 100 μm in size. Only <0.2% of the particles were between 100 μm and 150 μm. The number of particles per vial was 4,500,000. Freshly prepared sodium pertechnetate in a volume of 1–10 mL was added to a vial of TechneScan LyoMAA. The vial was carefully swirled a few times and incubated for 10 min at room temperature.

### 3.1. Preparation of ^99m^Tc-MAA with Omnipaque Contrast Media

Approximately 2–7 mCi of ^99m^Tc was added to six different vials containing 0.7–2.5 mL of macroaggregate generating ^99m^Tc-MAA. The sample vials were left to stand upright at room temperature for 15 min. Then, 3 to 5 mL of omnipaque was added to the vial containing ^99m^Tc-MAA immediately after the radiochemical purity measurements were finished. The omnipaque and ^99m^Tc-MAA mixture was allowed to stand at room temperature for 3.5 to 4 h. The radiochemical purities of the ^99m^Tc-MAA samples before addition of omnipaque into samples and after 3.5 to 4 h of standing with the addition of omnipaque were measured using iTLC with iTLC-strips as the stationary phase and normal saline as the mobile phase. The iTLC strips were cut into two halves, and the radioactivity in each half was determined using a multichannel analyzer calibrated daily with a ^137^cesium calibration source (Table 1). The ^99m^Tc-MAA was expected to be found in the lower half, while the unlabeled ^99m^Tc was expected to be found in the upper half of the strip.

### 3.2. Preparation of ^90^Y-Microsphere in Various Concentrations of Omnipaque in D5 or Saline Solution

Five samples of three different concentrations of omnipaque mixed with D5 or saline solution were prepared in different vials. The concentrations included 100%/0%, 75%/25% and 50%/50% of omnipaque and D5 or normal saline solution, respectively. A total volume of 5 mL was used for each sample. Approximately 4–7 mCi of ^90^Y-microspheres per vial were suspended into the five different sample vials of various concentrations. Then, the five samples in vials were left to stand upright and undisturbed at room temperature for 4 h to determine the radiochemical purities using iTLC. The radiochemical purities of the ^90^Y-microsphere samples were measured using iTLC with iTLC-silica gel (1 × 5cm) as the stationary phase and normal saline as the mobile phase. The radiochemical purities of the ^90^Y-microsphere samples were calculated on the basis of the integral areas of the two peaks of ^90^Y-microsphere and free ^90^Y, in the iTLC chromatogram. ^90^Y-microsphere experiments were repeated three times, and the average results are reported in Table 2.

### 3.3. Microscopic Imaging of ^90^Y-Microspheres

Five samples of various concentrations of omnipaque mixed with D5 or saline solution were prepared for imaging under the microscope. Approximately one drop (~200 uL) of samples was added onto a microscope slide using a pipette and covered with a cover glass before being imaged under the microscope. For comparison, after 3.5 to 4 h, another drop from each of the above five samples were taken for imaging under the microscope.

## 4. Conclusions

^99m^Tc-MAA and Y-90 microspheres can be safely suspended in a solution of omnipaque, which can improve the ability of the interventional radiologist to visualize the flow of particles in real time. Implications would be a safer injection of ^99m^Tc-MAA particles with better pre-therapy planning and on-table monitoring of the distribution of ^90^Y-microspheres.

## Figures and Tables

**Figure 1 molecules-27-07646-f001:**
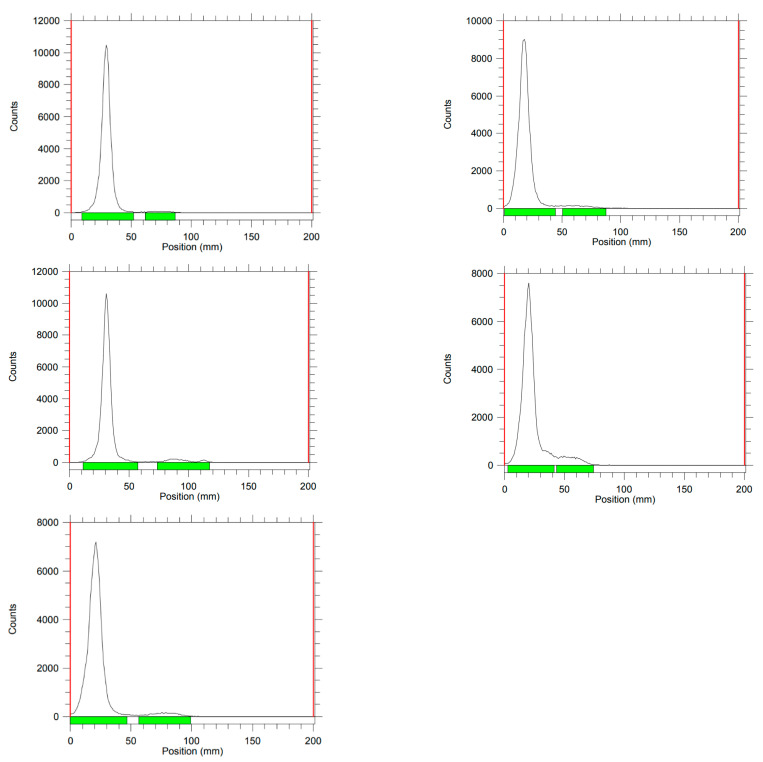
iTLC chromatograms of samples **1**–**5**.

**Figure 2 molecules-27-07646-f002:**
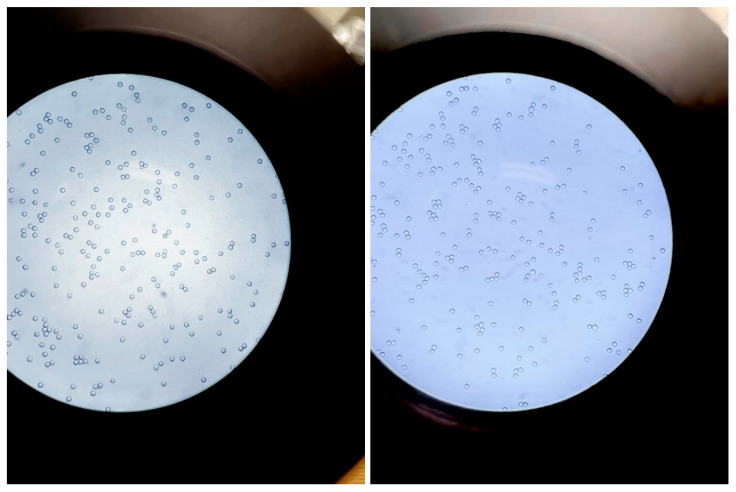
Microscopic images of samples **3** (**Left**) and **5** (**Right**). Microscopic images of **1**, **2** and **4** samples are provided in the Appendix A.

**Table 1 molecules-27-07646-t001:** (**a**) Volume of omnipaque, MAA, and ^99m^Tc radioactivity of six different samples. (**b**) Radiochemical purity as a percentage (%RCP) of the above samples before and after the addition of omnipaque.

Sample	Omnipaque (mL)		MAA (mL)	^99m^Tc-MAA (mCi)		Time Elapsed (Hrs)
**1**	3		0.7	5.30		3.5
**2**	5		1.2	2.94		4
**3**	5		1.5	6.67		4
**4**	5		1.2	7.32		4
**5**	5		2.1	6.55		4
**6**	5		2.5	6.28		4
	**Before Addition of Omnipaque**			**After Addition of Omnipaque**		
**Omnipaque (mL)**	**Lower Half**	**Upper Half**	**%RCP**	**Lower Half**	**Upper Half**	**%RCP**
3	263,315	130	99.95	18,478	255	98.64
5	138,511	250	99.82	17,484	340	98.09
5	93,462	174	99.81	56,710	567	99.01
5	125,548	284	99.77	10,764	204	98.14
5	130,513	178	99.86	35,848	405	98.88
5	101,388	239	99.76	48,381	181	99.63
		Average	99.83		Average	98.73
		SD	0.068		SD	0.578

**Table 2 molecules-27-07646-t002:** The radiochemical purity as a percentage (%RCP) at various concentrations of omnipaque and D5 or saline solution ratio in five different samples (**1**–**5**).

	1	2	3	4	5
Omnipaque conc and vol, D5 or Saline vol. added	100%, 5 mL	75%, 3.75 mL; D5, 1.25 mL	50%, 2.5 mL; D5, 2.5 mL	75%, 3.75 mL; saline, 1.25 mL	50%, 2.5 mL; saline, 2.5 mL
1	98.35%	94.67%	98.11%	90.02%	93.61%
2	97.59%	94.63%	96.35%	97.02%	97.08%
3	98.29%	98.17%	98.48%	97.85%	96.31%
Average	98.08%	95.82%	97.65%	94.96%	95.67%
SD	0.004	0.020	0.011	0.043	0.018

## Data Availability

No applicable.

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
