# Peer review of "Radiochemical Feasibility of Mixing of 99mTc-MAA and 90Y-Microspheres with Omnipaque Contrast"

_molecules, 2022, doi:10.3390/molecules27217646_

Round 1

Reviewer 1 Report

This is an ex vivo study of the compatibility and stability characteristics of 99mTc-MAA and 90Y microspheres, each suspended in various concentrations of a standard contrast agent (omnipaque). Currently, selective hepatic artery administration of 99mTc MAA and 90Y-microspheres is performed “blindly” because both are radiolucent. Results show that the radiochemical of mixing 99mTc MAA and 90Y-microspheres with omnipaque is acceptable, which would improve the ability of the interventional radiologist to visualize the flow of delivered particles in real time under fluoroscopy during selective internal radiation therapy (SIRT).

Even though further evaluation with relevant animal models for in vivo studies is needed, the present analysis is interesting and opens the possibility for future application in humans.

Major comments

- The abstract should include that radiochemical purity and feasibility of mixing 99mTc-MAA with various concentrations of a standard contrast agent was also investigated.

- 2.5. Image acquisition section (page 3) is irrelevant for the analysis performed, misleads the reader, and should be removed from the manuscript.

- Figures 3 and 4 are not related to the methods used nor are they relevant for the analysis performed, so they should be removed together with the related paragraphs in the body of the manuscript (including references 24-26).

Minor comments

- Since the contrast agent for 90Y-SIRT is administered intraarterial, the authors should consider removal of “intravenous” in the following sentences:

Introduction, page 2: “Injection of intravenous iodinated contrast into a microcatheter that is placed in the selected hepatic artery allows…”

Results and discussion, page 4: “A small bolus of intravenous contrast such as omnipaque was injected to allow delineation of the arterial anatomy…”

- Results and discussion section, page 4: consider the following verb time change in the sentence “contrast such as omnipaque is injected to allow delineation of the arterial anatomy”, instead of “contrast such as omnipaque was injected to allow delineation of the arterial anatomy”

- There is a typo in the 1st paragraph of Clinical applications of mixing of 99mTc-MAA and 90Y microspheres with omnipaque section, on page 7: “…90Y-microspheres to provide details…”, instead of “90Y-microspheres to provides details”.

Author Response

Reviewer 1.

This is an ex vivo study of the compatibility and stability characteristics of 99mTc-MAA and 90Y microspheres, each suspended in various concentrations of a standard contrast agent (omnipaque). Currently, selective hepatic artery administration of 99mTc MAA and 90Y-microspheres is performed “blindly” because both are radiolucent. Results show that the radiochemical of mixing 99mTc MAA and 90Y-microspheres with omnipaque is acceptable, which would improve the ability of the interventional radiologist to visualize the flow of delivered particles in real time under fluoroscopy during selective internal radiation therapy (SIRT).

Even though further evaluation with relevant animal models for in vivo studies is needed, the present analysis is interesting and opens the possibility for future application in humans.

Thank you very much for the positive comments.

Major comments

- The abstract should include that radiochemical purity and feasibility of mixing 99mTc-MAA with various concentrations of a standard contrast agent was also investigated.

Answer: Thanks for the comment. The statement has been included in the abstract. See manuscript p2.

- 2.5. Image acquisition section (page 3) is irrelevant for the analysis performed, misleads the reader, and should be removed from the manuscript.

- Figures 3 and 4 are not related to the methods used nor are they relevant for the analysis performed, so they should be removed together with the related paragraphs in the body of the manuscript (including references 24-26).

Answer: Thanks for the comments. Agreed. We have removed the Image acquisition section, Fig 3, Fig 4 and related paragraphs.

Minor comments

- Since the contrast agent for 90Y-SIRT is administered intraarterial, the authors should consider removal of “intravenous” in the following sentences:

Answer: Thanks for the comment. Agreed.

Introduction, page 2: “Injection of intravenous iodinated contrast into a microcatheter that is placed in the selected hepatic artery allows…”

Answer:  done , “intravenous” was removed.

Results and discussion, page 4: “A small bolus of intravenous contrast such as omnipaque was injected to allow delineation of the arterial anatomy…”

Answer: Done, “intravenous” was removed.

- Results and discussion section, page 4: consider the following verb time change in the sentence “contrast such as omnipaque is injected to allow delineation of the arterial anatomy”, instead of “contrast such as omnipaque was injected to allow delineation of the arterial anatomy”

Answer: Thanks for the comments. We have changed “was” to “is”. See p 7.

- There is a typo in the 1st paragraph of Clinical applications of mixing of 99mTc-MAA and 90Y microspheres with omnipaque section, on page 7: “…90Y-microspheres to provide details…”, instead of “90Y-microspheres to provides details”.

Answer: We have corrected the typo. But the session has been removed according to what you suggested.

Reviewer 2 Report

The authors have presented the feasibility of suspending 99mTc-MAA and 90Y-microspheres in omnipaque to improve the real time visualization of flow of particles. The authors have also validated their approach in clinical scenario using PET and SPECT imaging studies. 

Comments: 

The stability of 99mTc-MAA and 90Y-microspheres-omnipaque preparations were not referenced or presented (serum stability). The combination of these agents may increase the radiolysis, particularly of 99mTc-MAA. Authors should provide references or data to demonstrate that the combination of these agents do not increase radiolysis. 

Please not any indications and contraindications that may occur due to combining agents or alone.  Omnipaque use can cause blood clots (also mentioned in the manuscript). The authors should reference previous works (clinical and pre-clinical) or provide data to support the safety of combining these agents. 

Please not the group size utilized in the clinical studies, a N of 1 limit the validation of this approach.  

Author Response

Reviewer 2:

The authors have presented the feasibility of suspending 99mTc-MAA and 90Y-microspheres in omnipaque to improve the real time visualization of flow of particles. The authors have also validated their approach in clinical scenario using PET and SPECT imaging studies.

Thank you very much for positive comments.

Comments:

The stability of 99mTc-MAA and 90Y-microspheres-omnipaque preparations were not referenced or presented (serum stability). The combination of these agents may increase the radiolysis, particularly of 99mTc-MAA. Authors should provide references or data to demonstrate that the combination of these agents do not increase radiolysis.

Answer: Thanks for the comments. We have included this statement in the discussion: The combination of the 99mTc-MAA or 90Y-microspheres with omnipaque did not increase the radiolysis of 99mTc-MAA or 90Y-microspheres as radiolysis depends on the linear energy transfer (LET) of the radioisotopes [23]. After adding contrast omnipaque, the radiolysis should not increase. Moreover, 99mTc is low energy gamma particle, the radiolysis of 99mTc-MAA should be much lower than that of 90Y-microspheres. We have tested the stability of 99mTc-MAA adding contrast omnipaque in same volume of serum incubated at 37 oC. TLC chromatograms at 1 hour and 3 hours showed that 99mTc-MAA is stable with contrast in serum (supplementary materials Fig S2). Moreover, if radiolysis has occurred or in vivo break down, the free 99mTc would have localized in thyroid and stomach. This did not happen in our study. 

  1. Collinson, E.; Dainton, F. S.; Kroh, J. Effects of linear energy transfer on the radiolysis of water and heavy water. Nature. 1960, 187, 475–477.

Please not any indications and contraindications that may occur due to combining agents or alone.  Omnipaque use can cause blood clots (also mentioned in the manuscript). The authors should reference previous works (clinical and pre-clinical) or provide data to support the safety of combining these agents.

Answer: Thanks for the comment. We will include this statement in the discussion: The current standard of care ‘sandwich’ technique involves omnipaque injection just before and just after 99mTc-MAA or 90Y-spheres infusion, which is not noted to have an increased risk of complications such as blood clots. Thus, it would be unlikely that the combination of 99mTc-MAA or 90Y-microspheres with omnipaque would result in such complications and even if so, these are unlikely to be more than the current standard of care.

Please not the group size utilized in the clinical studies, a N of 1 limit the validation of this approach.

Answer: thanks for the comments. Agreed with reviewer. But according to the reviewer 1’s suggestion, we have removed that part.

Round 2

Reviewer 2 Report

Authors have incorporated the changes required. The manuscript is now acceptable in its present form.